# Hybrid Blockchain for IoT—Energy Analysis and Reward Plan

**DOI:** 10.3390/s21010305

**Published:** 2021-01-05

**Authors:** Jiejun Hu, Martin J. Reed, Mays Al-Naday, Nikolaos Thomos

**Affiliations:** School of Computer Science and Electronic Engineering, University of Essex, Colchester CO4 3SQ, UK; jiejun.hu@essex.ac.uk (J.H.); mfhaln@essex.ac.uk (M.A.-N.); nthomos@essex.ac.uk (N.T.)

**Keywords:** hybrid blockchain, energy evaluation, reward plan

## Abstract

Blockchain technology has brought significant advantages for security and trustworthiness, in particular for Internet of Things (IoT) applications where there are multiple organisations that need to verify data and ensure security of shared smart contracts. Blockchain technology offers security features by means of consensus mechanisms; two key consensus mechanisms are, Proof of Work (PoW) and Practical Byzantine Fault Tolerance (PBFT). While the PoW based mechanism is computationally intensive, due to the puzzle solving, the PBFT consensus mechanism is communication intensive due to the all-to-all messages; thereby, both may result in high energy consumption and, hence, there is a trade-off between the computation and the communication energy costs. In this paper, we propose a hybrid-blockchain (H-chain) framework appropriate for scenarios where multiple organizations exist and where the framework enables private transaction verification and public transaction sharing and audit, according to application needs. In particular, we study the energy consumption of the hybrid consensus mechanisms in H-chain. Moreover, this paper proposes a reward plan to incentivize the blockchain agents so that they make contributions to the H-chain while also considering the energy consumption. While the work is generally applicable to IoT applications, the paper illustrates the framework in a scenario which secures an IoT application connected using a software defined network (SDN). The evaluation results first provide a method to balance the public and private parts of the H-chain deployment according to network conditions, computation capability, verification complexity, among other parameters. The simulation results demonstrate that the reward plan can incentivize the blockchain agents to contribute to the H-chain considering the energy consumption of the hybrid consensus mechanism, this enables the proposed H-chain to achieve optimal social welfare.

## 1. Introduction

The Internet of Things (IoT) is a rapidly developing field with the number of Internet Protocol (IP) devices connected to the Internet predicted to be three times the global population by 2023 [1]. A large number of IoT applications cross organisational boundaries, from device owner, network provider, application framework and cloud provision. For example, an intelligent transport system (ITS) requires sensors in vehicles, owned by individuals, to interact with roadside units, managed by the ITS provider, who uses a network operator to interconnect their systems with cloud provision to host analytics [2]. It is essential that these organizations can inter-operate in an efficient, secure and trustworthy manner. While many technologies are required to enable this cooperation, this paper concentrates on how blockchain technologies can provide shared data or *contracts* in a manner that allows for both intra-organization and inter-organization blockchain systems. This is achieved by a hybrid-blockchain (H-chain) that takes advantages of combining blockchain systems that suit either intra and inter-organisation into a unified H-chain.

While a number of blockchain systems exist, two common approaches are—proof of work (PoW) based consensus mechanisms and Practical Byzantine Fault Tolerance (PBFT) consensus mechanisms [3]. Blockchains based on a PoW consensus mechanism are computationally-intensive and hence energy-expensive [4], however, they provide excellent trustworthiness in a system that spans organizational boundaries. On the other hand, a PBFT based consensus mechanism is communication-intensive [3] but has been widely used as a permissioned private blockchain system. Thus, there is a motivation for a combination of these systems to achieve a balance between verification performance and energy cost.

IoT applications may require a broad range of information from multiple organisations to collaborate and provide a more powerful service to the users. An example of a multi-organizational IoT ecosystem has been demonstrated by the project “Secure and safe Internet of Things” (SerIoT) [5] which uses software defined networking (SDN) to assist IoT applications in delivering an IoT security application; this will be one of the example use cases within this paper. In any multi-party application, the collaboration between multiple organisations requires that the boundaries of information are clearly delineated. Particularly, there exists three types of information which can be known by an organisation: (a) information private to each organisation, such as IoT users’ private data or device logs that should only be verified and shared within the organisation; (b) public information, like shared databases of malicious behaviour and software integrity information, that needs to be circulated among organisations; (c) hybrid information that are only required by limited number of organisations, for example, when organisations form a partnership with shared information. These various types of information in the IoT application render a purely private blockchain insufficient, which drives us to design a more flexible blockchain solution.

Consequently, depending upon the type of information (private/shared/public), a combination of both public private blockchains are needed to facilitate all the application requirements. There are four scenarios we illustrate in Figure 1a with examples to facilitate the descriptions:Scenario 1—Private chain within a single organisation: In this scenario, Organisation 1 has private transactions, such as sensory data generated by local IoT application, to be verified and stored within the organisation. As shows in Figure 1a, sensory data of Organisation 1 can be verified by organisation-owned servers to preserve the privacy.Scenario 2—Private chain across organisations: Organisation 1 and 2 form a partnership for an IoT application. The two organisations share IoT devices, and the IoT devices or systems communicate with each other (in Figure 1a). For example, in the aforementioned SerIoT IoT security application [5], after the SDN controller’s path calculation, the flow rules are verified across organisations 1 and 2 by PBFT consensus mechanism.Scenario 3—Public chain and private chain cooperation: if, after the flow rule verification by private chain, malicious behaviour is detected, then, the organisations 1 and 2 decide to make this information public to all the organisations as an alert to block a certain system (as in Figure 1b). In such a case, the leading agent in the private blockchain initiates a public chain consensus mechanism, which requires communication between each organisation. This scenario enables a block/allow list to be made public.Scenario 4—Public chain only: Organisation 1 for example may notice one of its IoT devices is suffering a denial of service attack from a specific IP address. Instantly, organisation 1 sends this information/transactions directly to the public chain, which informs all the organisations (as in Figure 1b).

These scenarios illustrate the earlier stated motivation for the H-chain. While the concept of a hybrid-blockchain has been suggested before [3,6,7,8,9], this paper gives greater depth to the concept with analysis that compares the performance of the two systems to allow an appropriate trade-off between performance and energy cost to be considered. A private blockchain, for example, Hyperledger [10], is a special type of blockchain that is *permissioned*. Such a private blockchain is able to support full privacy of the chain owner and high veracity verification through the PBFT based consensus mechanism. However, its efficiency degrades as the private blockchain network increases in size mainly, due to the communication intensive PBFT consensus. A public blockchain, for example, Ethereum [11], provides a generic solution offering decentralisation, scalability, and public access. Public blockchain solutions adopt Proof of Work (PoW) to reach consensus among all the participants. PoW requires the participants to join the competition of puzzle solving, which is computation intensive. In this work, we propose H-chain that balances transaction efficiency and the scalability from combining the private and public solutions.

Edge, fog and cloud computing provide IoT applications with the flexibility to deploy energy/computation intensive technologies, that is, blockchain. This reduces the energy cost of the IoT devices which are battery limited. However, the energy consumption still remains an important issue, as the energy cost has simply shifted from the IoT devices to the edge computing servers. Blockchain technology is energy intensive. For example, it is estimated that the annual total footprint of Bitcoin mining is comparable to the carbon footprint of New Zealand [12]. This fact has violated the Paris Agreement climate change commitments that technology should be utilised to achieve greenhouse gas mitigation [13]. Sustainability is crucial in the design and deployment of blockchain technology.

Thus, in this paper, we first propose H-chain to replace a pure PoW solution with a permissioned-PoW and PBFT combination. This also enables private transaction verification and public transaction sharing/audit. Then, we study the energy consumption of the permissioned-PoW, PBFT and H-chain, respectively. Last but not least, we propose a reward plan to compensate the energy cost of the H-chain under limited reward budget. The reward plan aims to encourage the number of verifiers in the private chain and the resource contribution of the miners in the public chain.

To our knowledge, this is the first work that proposes hybrid-blockchain to support cooperation between multiple organisations with a reward plan. Our previous work [14] has focused on SDN flow verification of a single organisation supported by private blockchain technology. An SDN-based IoT scenario is one of the use cases of the proposed H-chain, for example to support secure flow negotiation in the SerIoT application [5]. H-chain aims to enable a flexible consensus mechanism and energy consumption based reward plan. The main contributions of this paper are summarised as follows.

we introduce the architecture of H-chain in this work to provide a customised consensus mechanism, which includes permissioned-PoW and PBFT;we analyse the energy consumption of both permissioned-PoW and PBFT consensus mechanism in the proposed network architecture. In particular, we extensively evaluate how the key factors, such as network conditions, computation capability, the number of organisations, and the number of blockchain agents, of the H-chain affect energy consumption.we study the design of the proposed reward plan to compensate the energy cost of the verifiers and the miners. The reward aims to encourage the use of more verifiers in the private chain and greater resource contribution by the miners. To provide guideline for H-chain, we consider the proportion of the blocks that stay private as a key factor in the reward plan.

In the following, we first discuss the related works in Section 2. Then, we propose the architecture of the H-chain and its advantages in Section 4. Next, we provide the system model and analyse the workflow of the permissioned-PoW and PBFT in Section 5. In Section 6, the reward plan to stimulate the blockchain agents is designed. Our solution is simulated extensively in Section 7 to get a better understanding of the parameters in the H-chain. Finally, we draw conclusions in Section 8.

## 2. Related Works

In this section, we review existing propositions to develop hybrid blockchain solutions supporting various IoT-based scenarios [3,6,8,9] and examine some of the energy-related challenges associated with the use of blockchain technologies.

### 2.1. Hybrid Blockchain

Desai et al. [15] proposed one of the earliest research systems combining public blockchain and private blockchain. Their solution leverages the use of private blockchain to open sensitive bids to auctioneers only, while public blockchain is used to announce the auction winner and to account the corresponding payment. The solution further offers a thorough definition of smart contracts for bidding, enabling fraud detection and orchestrate the auctioning process. However, their assessment lacks analytical and quantitative evaluation of the proposed hybrid approach. We feel that an analysis of the tradeoff between the two approaches is needed, especially given the urgency of addressing the energy consumption in blockchain. Zhu et al. [8] presented a hybrid blockchain-based crowdsourcing platform and utilised Delegated Proof of Stake (DPoS) and PBFT consensus mechanisms to enable efficient transaction verification. The authors compared the throughput of each mechanisms and concluded that DPoS has greater throughput compared to a PoW consensus, however, again a trade-off between consensus mechanisms and the energy consumption was not addressed. Yazdinejad et al. [16] proposed an energy-efficient IoT network with blockchain-based security solution. They presented a SDN-based cluster architecture where there is a SDN controller as cluster head in each cluster, i.e., SDN domain. In that work, they proposed an intra-domain private blockchain and inter-domain public blockchain that enables flexibility of IoT device migration. However, there was no analysis about energy consumption. In addition, classic PoW was not introduced due to the consideration of energy efficiency. The works in References [17,18] presented energy efficient blockchain by replacing PoW and bitcoin out of the verification into distributed trust in IoT based networks. However, the work did not describe how miners would interact with each other. In addition, there was no detailed modeling of energy consumption related to blockchain. Replacing PoW is one of the solutions for energy preservation, however, it also reduces the security. Thus, modeling energy consumption would make the trade-off in terms of security possible. Kim et al. [19] systematically reviewed scientific papers and industrial white papers, and then introduced the architecture, connectivity, interoperation of heterogeneous blockchains. However, the discussed works on hybrid blockchain are mainly focused on presenting the architecture of public and private blockchain and the business logic deployed by the smart contracts. There is limited work, such as that of Reference [20], that provide performance evaluation of the hybrid blockchain. Sagirlar et al. [20] first investigated the performance of the PoW in blockchain-IoT with respect to the block generation intervals, device locations, and the number of peers. Second, a hybrid blockchain that includes the BFT-based inter-domain consensus and the PoW-based intra-domain consensus was introduced.

Besides hybrid blockchain, there is also a concept called *consortium blockchain* [21]. A consortium blockchain is also managed by multiple organisations, in which the user nodes are authorised as private and public nodes. A consortium blockchain is governed by a group and not by a single entity. Conversely, hybrid blockchain has more flexibility and scalability than consortium blockchain, since first the user nodes can be either in the public chain or the private chain, second transactions are verified first by the private chain then by the public chain without compromising privacy.

Differently from the above mentioned works, in this work, we analyse the energy consumption of the proposed H-chain by considering a number of parameters, such as the network conditions, computation capability, the number of organisations, and the number of blockchain agents. The extensive evaluation contained in this paper provide insights that can can serve as a guideline of hybrid blockchain deployment.

### 2.2. Energy Consumption of Blockchain

Blockchain is resource exhausting technology, particularly PoW based systems such as Bitcoin [22] and Ethereum [23]. This contradicts with the limited energy budget in IoT devices. Hence, when deploying blockchain in IoT application, energy efficiency is one of the most important issues. There are only a few works studying the energy consumption of hybrid blockchain. Specifically, Reference [16] proposed energy efficient hybrid blockchain assisted IoT networks by presenting a novel cluster-based routing protocol. However, this work did not propose theoretical analysis and optimise the energy consumption. Sedlmeir et al. [24] thoroughly studied the PoW consensus mechanism, and in particular the upper and lower bound. They, also, argued that PoW cryptocurrencies are not likely to become a major threat to the climate in the future. Reference [25] investigated the energy consumption of different PoW based cryptocurrencies. Sharma et al. [26] presented an energy-efficient transaction model for the blockchain-enabled Internet of Vehicles by optimising the number of transaction offloading. Reference [27] proposed modified PoW that includes two stages, which can reduce energy consumption. It is widely known that PoW based consensus mechanism is computational intensive, and PBFT based consensus mechanism is communication intensive. The aforementioned works only concentrated on the energy cost of one of the consensus mechanisms and, thus, are not able to analyze energy cost of the proposed H-chain which combines both consensus mechanisms. Our scheme provides a mechanism to trade off the communication and computation complexity according to customised consensus mechanism defining the proportion of the private and public blocks. This is discussed in detail in Section 5 and Section 6.

## 3. Advantages of H-Chain

The earlier arguments have given the motivation for introducing H-chain. Next, we specifically describe the benefits of H-chain to a multi-organisation IoT scenario as follows:Selective information exposure: The organisations can keep the privacy of their own data and define complex verification contracts within organisations.PoW-level of security without PoW work across all transactions: after verification by the private chain, the public chain only needs to verify the hash of a transaction, which leads to more efficient public blockchain, since in pure PoW the whole block and transactions need to be inspected. Moreover, in this work, we propose a permissioned-PoW that enables PoW to run on the permissioned devices verified by the private blockchain.Multiple chain security: Some of the transactions have to go through verification of both the private chain and the public chain verification. This process not only provides multiple layers security, but also removes the transaction congestion on the public chain as some transactions are shifted to the private chain.Reduce risk of attack on transactions: Some of the organisation-owned information is private, which leads to an unpredictable block generation rate. This fact makes it hard for the attacker to carry out malicious behaviour compared to the public blockchain where block creation is known.

The advantages of a hybrid blockchain are not just limited to the headline advantages given above. For example, a hybrid blockchain operates in a closed ecosystem; that is, each organisation grants permission to the IoT devices and the servers, and in addition, organisations have mutual consensus when forming a partnership. This not only enhances security, but also protects the privacy while organisations still communicate with the outer world. Additionally, organisations can decide the proportion of effort attributed to the private or public chain depending upon the privacy of the data and depending upon energy/performance criteria. These features enhance the flexibility and scalability of a blockchain based IoT application using H-chain.

## 4. Architecture of Hybrid-Blockchain

We consider a multiple-organisation scenario where one organisation can interact with one or more other organisations. These organisations are connected via a network that could be private or through public peered networks and the organisations have their own IoT applications and business model. In Figure 1a, we imagine a scenario where the organisations are using a SDN network which is one use case we are considering, but this is not restrictive. As expected, the IoT applications and business models of each organisation are private information to each organisation. However, we assume that the organisations require information verification, sharing, and audit with each other. Examples of this shared information include applications such as: network operation/management, malicious behaviour history, SDN flow rule management (e.g., as requested “intents”), external routing reachability to name just a few. Moreover, H-chain introduces the combination of organisation-owned private blockchain and public blockchain that enables private information verification and public information sharing as required.

### Entities and Structure of H-Chain

Organisations and service providers are now moving towards flexible network architectures with organisation-managed computation resource spanning different physical domains, that is, local servers and edge/fog/cloud computing infrastructure, that facilitate IoT applications and data storage requirements. The proposed H-chain utilises the computing resource of the organisation to assist the organisation-owned private blockchain and the public blockchain. Below, we list the important entities of H-chain as indicated in Figure 1a. While we use the example of an SDN security application to verify network flows, the general architecture can be used for any data types which might require private, public or shared verification, depending upon the specific requirements of each data item.

*Blockchain agent (BCA)*: are software components (i.e., servers) utilising edge computing. BCAs are in charge of the flow verification/validation (and other information) via smart contracts. Furthermore, BCAs also execute basic blockchain functions, such as the consensus process, sending transactions, and maintaining the flow ledger. We assume that for each organisation, it requires at least three BCAs (to fulfill the requirement of PBFT) to form the private blockchain.*Leading BCA*: there is one *leading* BCA in each organisation that is part of the public blockchain as well as a BCA as explained above. Every leading BCA is able to communicate with other leading BCAs and coordinates not only permissioned-PoW for the public chain, but also PBFT for two or more organisations’ private chain. The leading BCA, namely the miner, can recruit the rest of the BCAs in the organisation to contribute to the public chain.*Private chain*: is owned by an organisation or a group of them in partnership. Private blockchain is in charge of private information verification, which is aided by the PBFT-based consensus mechanism. More than three BCAs are required to operate the private chain;*Public chain*: is operated by the leading BCAs of each organisation. Permissioned-PoW is adopted in public chain for public information verification, validation, storing, and audition. In this work, we propose permissioned-PoW that is similar to traditional PoW, with the difference that it has permissioned miners, that is, leading BCAs. We use PoW as the consensus mechanism of the public chain in the rest of the work.*Connectivity*: intra-organisation connectivity among BCAs is facilitated through an internal network; in our use-case example this is through SDN. Inter-organisation connections are enabled either by dedicated links owned by the connected organisations or provided by a third party—for example, a national or international-network provider.

## 5. System Model

In this section, we first introduce the basic transmission and computation model of the PoW and PBFT based blockchains. Then, we investigate the energy consumption of these consensus mechanisms according to the workflow. Let us define an organisation with index i∈{1,2,...,I} and the number of the BCAs within it as $N_{i}$. For simplicity, without loss of generality, we assume an equal number of BCAs within each organisation. We consider the following SDN IoT scenario as an example, but it could equally apply to any IoT application which requires validation of some process or data. When a new communication packet is sent by a sensor, the corresponding switch will forward this packet to the SDN controller to obtain the appropriate flow rule. Then, the new flow rule will be forwarded to the leading BCA, where the verification process is triggered. The leading BCA gathers the transactions and packetises them into a block. The consensus processing of the block depends upon the operational requirements (e.g., inter/intra organisation). We define the block size and the number of transactions in one block as *s* and *K* respectively. We define the intra-organisation and inter-organisation effective throughput as *R* and $R_{c}$, respectively, and the size of acknowledgement message as $s_{ack}$. For BCAs, the CPU capability, in terms of number of performed operations per-second, is denoted as *f*. Table 1 provides a summary of our notations.

### 5.1. Permissioned Proof-of-Work and Energy Modeling

In our H-chain solution, the leading BCA in each organisation is in charge of public information verification. Our earlier work [28] introduced a single-organisation, PBFT-based, workflow in which leading BCAs run a permissioned-PoW once there is a new block. In this paper, we extend this approach to both PBFT- and PoW-based consensus, applicable across not only single but also multiple organisations. To this end, the extended PoW-based workflow we are proposing for multiple organisations is described as below (the extended PBFT-based workflow is described in the next section). Again we use the scenario of an IoT scenario with secured SDN to illustrate the workflow, but it could equally apply to any data/process verification. The workflow proceeds as follows:1.Leading BCAs collect the new data, for example, SDN flow rule [28], that is ready for verification, and build a block. We define the computational latency of this step as $C_{1}\left(s\right)$, where the latency is proportional to the blocksize *s*. Note that, the leading BCA in each organisation is also the miner of the public chain.2.According to the application requirements, that is, types of knowledge, business model, and energy efficiency, the leading BCA decides the proportion of the blocks that goes public during a period of time. In Section 6, we propose the reward plan for miners in respect to the proportion of the blocks.3.All the miners begin to solve the PoW puzzle. The winning miner’s PoW computation latency is denoted as $C_{2}$, and the rest of the miners’ computational latency is $C_{2}^{\prime }>C_{2}$.4.The winning miner completes the PoW and broadcasts the new block to all the miners. Here, the resulting transmission latency is defined as $T_{c}$, and it is dependent on the number of organisations/miners *I*. If the first generated block is delayed during transmission, then the miners may mistake the second block as the first one. This phenomenon is termed *forking*. For simplicity, without loss of generality, we assume there is no forking in this work.5.The other miners receive the new block, stop the current PoW, verify the data in the new block and if it passes, then accept and append the new block. The computational latency of new data verification and appending the new block is denoted as $C_{3}$. Till here, the miner of each organisation begins to disseminate the new block inside the organisation. The intra-organisation dissemination latency is defined as $T_{b}\left(s\right)$, which is related to the blocksize.

We denote the power required for transmission as $P_{t}$ and the computational power as $P_{c}$. The energy consumption is
(1)E=P·T,
where *P* is power (in Watt), that is, transmission power $P_{t}$ and computational power $P_{c}$, and *T* is latency (in second) thus expressing energy in standard units of Joule(watt/s). Based on the workflow described above, the energy consumption of PoW can be computed as:$$E_{PoW}=P_{t}\left(T_{b}\left(s\right)+T_{c}\left(I\right)\right)+P_{c}\left[C_{1}\left(s\right)+C_{2}+C_{2}^{\prime }+C_{3}\right].$$
When information is transmitted, there is some additional transmission latency and propagation latency, where the transmission latency is related to the communication link rate, and the propagation delay is proportional to the length of the link [29]. Note, we assume that the propagation delay within one organisation is negligible. In addition, we assume there is extra transmission latency across organisations that includes both actual media propagation delay and intermediary equipment such as switches and amplifiers of the links. We define the inter-organisation extra transmission cost factor as $\beta (T_{p})$, where this factor is proportional to the inter-organisation propagation delay $T_{p}$ reflecting the simplified assumption that transmission latency due to these additional components is related to distance. Thus, we have the transmission latency of the PoW
$$T_{1}\left(s,R,R_{c},I,T_{p}\right)=T_{b}\left(s,R\right)+T_{c}\left(s,R_{c},I\right)+\beta \left(T_{p}\right),$$
where we have the intra-organisation new block broadcast latency $T_{b}=\frac{s}{R}$ and the inter-organisation transmission latency $T_{c}=I\frac{s}{R_{c}}$.

The computational latency of the PoW, is strongly related to the difficulty of solving the PoW puzzle. Specifically, The difficulty of the PoW is defined as the number of the zeros in front of the hash value. With increasing number of the zeros, the difficulty of the PoW increases. We denote the difficulty factor as *D* that is related to the winning mining task’s size $\gamma^{*}$
(2)$$\gamma^{*}=B^{\kappa D},$$
where *B* is the basic puzzle size when there is one zero in front of the hash value, and κ∈(0,1) is the coefficient corresponding to the difficulty factor. Furthermore, we can define the non-winning miners PoW puzzle size in the same approach. Note, the non-winning miner *i* has puzzle size $\gamma_{i}\in \left(0,\gamma^{*}\right)$ (there is only one miner in one organisation, so we use *i* without losing generality), which follows normal distribution $\left(\mu ,\sigma^{2}\right)$, where μ is the mean value and σ is the variance, with probability $p_{i}$. The computation latency of building the new block is $C_{1}=\frac{s}{f}$. The puzzle solving latency of the winning miner and non-winners *i* is $C_{2}=\frac{\gamma^{*}}{f}$ and $C_{2}^{*}=\frac{\gamma_{i}p_{i}}{f}$, respectively. The latency of the transactions verification in the new block and the appending of the new block are combined into a single term $C_{3}=\frac{\gamma^{\prime }K}{f}$, where $\gamma^{\prime }$ is the complexity of the transaction verification. Then, the energy consumption is defined as
(3)$$E_{PoW}=P_{t}\left[\frac{s}{R}+I\frac{s}{R_{c}}+\beta \left(T_{p}\right)\right]+\frac{P_{c}}{f}\left(s+\gamma^{*}+I\sum_{i\neq i^{*}}^{N}p_{i}\gamma_{i}+IN\gamma^{\prime }K\right).$$

### 5.2. Practical Byzantine Fault Tolerance and Energy Modeling

PBFT in H-chain can be deployed across multiple organisations or within a single organisation; we will generalise the solution by formulating across multiple organisations unless stated otherwise. We present the workflow of PBFT in H-chain as below:1.The initiating BCA collects the new data, for example, SDN flow rules, as transactions and builds a block. Similar to PoW case we define the computational latency for this stage as $C_{1}$.2.According to an organisation’s requirement, the initiating BCA defines the number of following BCAs $N_{i}$ and the number of organisations *I*, and the proportion of the private blocks (details in Section 6). Then, the initiating BCA sends the new block to the other BCAs.
*2.1.* if the consensus is within one organisation, the new block is broadcast to all the following BCAs within the organisation. This leads to the broadcast transmission latency $T_{b}\left(s\right)$.*2.2.* if the PBFT is across organisations, there is an extra inter-organisation transmission latency accounting the latency introduced because of leading BCAs communication within each organisation. This is denoted as $T_{c}\left(s,I\right)$.3.The following BCAs first send all-to-all acknowledgement (ACK) messages to confirm the acceptance of the new block, which results in intra-organisation transmission latency $T_{a}^{\prime }$ and inter-organisation transmission latency $T_{c}^{\prime }$. Then, all the BCAs begin with the verification that incurs a verification latency $C_{2}^{\prime }$. For example, in our SDN/IoT scenario, BCAs conduct verification of the new SDN-flow as new data according to pre-defined flow conformance policy [28].4.Another all-to-all ACK messages exchanging happens to confirm the verification result, which is similar to Step 3 ($T_{a}^{\prime }$ and $T_{c}^{\prime }$).5.The initiating BCA waits for all the ACK messages from the following BCAs. If the votes reach the requirement, the BCAs append the new block to the ledger. In case the votes are inadequate for the consensus requirement, the initiator BCA is informed.

We first denote the size of ACK message as $s_{ack}$. Thus, the all-to-all intra-organisation transmission latency is given by $T_{a}\left(s_{ack},N,R\right)=\frac{s_{ack}}{R}N^{2}$ and the inter-organisation ACK transmission latency by $T_{c}^{\prime }=I^{2}\frac{s_{ack}}{R_{c}}+\beta \left(T_{p}\right)$. The energy consumption of the PBFT, $E_{PBFT}$, consists of communication and computation energy cost, where the communication energy includes the intra- and inter-organisation new block dissemination and twice all-to-all confirmation; The computation energy includes new block establish and verification. Thus, we have the energy consumption of one new block with PBFT as
(4)$$E_{PBFT}=P_{t}\left[\frac{s}{R}+I\frac{s}{R_{c}}+\beta \left(T_{p}\right)+2\left(\frac{s_{ack}}{R}N^{2}+\frac{s_{ack}}{R_{c}}I^{2}+\beta \left(T_{p}\right)\right)\right]+\frac{P_{c}}{f}\left(s+IN\gamma^{\prime }K\right).$$
As we can observe from the above equation, PBFT is communication intensive due to the all-to-all communication of the BCAs to confirm the consensus, that is, the quadratic form of the number of the BCAs and the number of the organisations.

### 5.3. Hybrid Blockchain and Energy Consumption

Organisations can benefit from the flexibility offered by our H-chain solution in deciding the proportion of the blocks to utilise the public blockchain for visibility as opposite to the proportion of the blocks that should stay private in one or more organisations. The decision on said proportions is taken by the respective organisation, once sufficient number of blocks is collected and ready for verification. The workflow of the H-chain follows the private chain and the public chain with coordination of the leading BCA. The leading BCA is aware of the verification requirement, and then initialises the consensus.

Thus, the energy consumption of the H-chain scenario is related to the private chain, public chain, and the proportion of the private blocks. To stimulate BCAs to make a contribution towards to the H-chain, each organisation has a reward budget (it can be monetary or reputation reward) for the transaction verification. The budget enables the organisation to choose the optimal number of BCA verifiers in the private chain and also stimulate the miners of the public chain to make the optimal contribution. Notably, although earlier we specify a minimum of three BCAs in each organisation, in reality their number could be significantly larger than three and hence the organisation would need to make an optimised selection. We consider that there is only one miner in each organisation, and that this is also the leading BCA. Since there would be multiple BCAs in one organisation, it is possible that the leading BCA recruits part of the rest BCAs in the organisation to contribute more resource, that is, computation capability. In this paper, we design a reward plan to incentivize the BCA verifiers and the miners in H-chain. We propose the optimisation problem presented in the Section 6, which aims to maximise the satisfaction of the verification initiating organisation.

## 6. Reward Scheme of H-Chain

In this section, we introduce the reward plan that considers H-chain energy consumption to stimulate the contribution of the BCAs. For the private chain, it is crucial to have multiple BCAs to join the verification to preserve the validity of the verification process. For the public chain, the leading BCA of each organisation has all the computational resource within the organisation to utilise and control, thus, we are interested in the resource the mining winner will utilise when tackling the puzzle of PoW. To model the problem, we first define the satisfaction function for both the private and public chain in terms of the reward and the energy consumption. We define the block generation rate of H-chain as ε. Hence, in a time period *l*, there would be εl blocks generated from one organisation. The organisation sets the proportion of blocks to be verified by private chain, according to the needs of the application, as ϕ∈(0,1), which means that there are ϕεl private blocks and (1−ϕ)εl public blocks.

### 6.1. Satisfaction Function of the Private Chain

We define the satisfaction function of the private chain as the profit, which is the income from the reward minus the cost of the energy. The private chain gains reward that is proportional to the number of verifiers in the private chain. We define the reward for each verifier as $r_{1}$. We assume that when an organisation raises a private block verification, it determines the number of organisations *I* according to the verification requirements. We denote the price of energy as η to balance the unit. According to the energy consumption of the private chain in (Equation 4), the satisfaction function of the private chain is
(5)$$\begin{array}{cc} U(N) & =\phi \varepsilon l[r_{1}N-\eta E_{PBFT}]\\ \to U(N) & =\phi \varepsilon l[r_{1}N-\eta \left(C+cN^{2}+dN\right)], \end{array}$$
where $C=P_{t}\left[\frac{s}{R}+I\frac{s}{R_{c}}+\beta \left(T_{p}\right)+2\left(\frac{s_{ack}}{R_{c}}I^{2}+\beta \left(T_{p}\right)\right)\right]+\frac{P_{c}}{f}s$, $c=\frac{s_{ack}}{R}$, and $d=\frac{P_{c}}{f}I\gamma^{\prime }K$. We want to maximise the satisfaction of the private chain, and require first U(N)≥0 in (Equation 5) and second N>0. As we observe, the utility function is concave, since there is a sum of a linear function and a quadratic function. This means that there is an optimal number of the BCA verifiers that should be use in the private blockchain.

### 6.2. Satisfaction Function of the Public Chain Bcas

For the public chain, we know that there is one miner (leading BCA) for each organisation in charge of the public chain. Thus, we know there is a definitive number of the leading BCAs in the public chain, which also equals to the number of the organisations. The leading BCA aims to finish the PoW puzzle and mine the block successfully to obtain the reward by providing more computation resource. Thus, the more computation resource will lead to higher probability of being a winning miner. We define the winning miner’s resource as $x_{l}$, and the resource of the miners *i* as $x_{i}$. The probability of being a winning miner is defined as
(6)$$p_{l}=\frac{x_{l}}{\sum_{i}^{I}x_{i}}.$$

The probability of being a winning miner is proportional to the resource that a miner puts into the mining [30]. Thus, (Equation 6) indicates that if a miner utilises more resource in the mining, the higher probability it is to be a winning miner. We assume that the total amount of the resource is known as $Z=\sum_{i}^{I}x_{i}$. Thus, we have $p=\frac{x}{Z}$.

We define the reward for the winning miner and the sum reward for the rest of the miners as $r_{2}$ and $r_{2}^{\prime }$, where $r_{2}^{\prime }=\xi r_{2},\;\xi >0$. We take the cost of PoW as a whole, that is, the expected energy cost includes the expected winning miner’s cost and the expected energy cost of the rest of the miners. The reward is allocated to the winning miner and all the rest of the miners according to the computation resource. For the energy cost of the PoW, the block transmission cost and the transaction verification cost remain the same, which we denote as $A=P_{t}\left[\frac{s}{R}+I\frac{s}{R_{c}}+\beta \left(T_{p}\right)\right]+\frac{P_{c}}{f}\left(s+I\gamma^{\prime }K\right)$. For the energy cost of solving the puzzle, it is obvious that it is proportional to the resource miners utilise. We denote *e* as the power factor of the resource. Thus, we have the energy cost of successfully solving the puzzle and the rest of the miners as ex and eZ, respectively. We now propose the satisfaction function of the public chain as
(7)$$\begin{array}{cc} U(x) & =\left(1-\phi \right)\varepsilon l\left[r_{2}x+\xi r_{2}Z-\eta E_{PoW}\right]\\ \to U(x) & =\left(1-\phi \right)\varepsilon l[r_{2}x+\xi r_{2}Z-\eta \left(A+pex+\left(1-p\right)eZ\right)]\\ \to U(x) & =\left(1-\phi \right)\varepsilon l[r_{2}x+\xi r_{2}Z-\eta \left(A+\frac{e}{Z}x^{2}-ex+eZ\right)]. \end{array}$$

To maximise the satisfaction function of the public chain, we require first U(x)≥0 in (Equation 7) and second x>0. As the utility function of the PoW in (Equation 7) is concave, there is an optimal resource contribution of the winning miner. Note that, we focus on the reward to the winning miner based on the computational resource it uses in mining. For the rest of the miners, we obtain the total reward as $\xi r_{2}$, which is proportional to the winning miner’s reward. This simple solution of reward allocation would be equally distributed.

### 6.3. Social Welfare Maximisation

In this section, we present the joint satisfaction of the H-chain by introducing the concept of social welfare. Social welfare is a widely used concept in economics [31]. We use it to interpret the economic efficiency and reward distribution in joint form. In H-chain, we aim to maximise the social welfare by allocating reward to the BCAs. By combining the satisfaction functions of the PBFT in (Equation 5) and the PoW in (Equation 7) together, we have the social welfare
(8)$$SW\left(N,x\right)=\phi \varepsilon l\left[r_{1}N-\eta \left(C+cN^{2}+dN\right)\right]+\left(1-\phi \right)\varepsilon l\left[r_{2}x+\xi r_{2}Z-\eta \left(A+\frac{e}{Z}x^{2}-ex+eZ\right)\right].$$

We aim to maximise the social welfare by finding the optimal number of the BCA verifiers and the optimal contribution of the winning miner under the constraint of the reward budget. Thus, we formulate the following maximisation problem
(9a)maxx,N SW(N,x)(9b)s.t. r1ϕεlN+(1−ϕ)εl(r2x+ξr2Z)≤Rb,(9c)U(N),U(x)≥0,(9d)N,x>0.

We should emphasize that the objective function (9a), includes both of the satisfaction functions of the PBFT and the PoW. The constraint in (9b) requires the total reward to be equal to the budget $R_{b}$. Problem (9a) can be solved by following the method of Lagrangian relaxation [32]. The constraint (9c) requires that the satisfaction functions are positive, and so to optimal variables in (9d). We define the Lagrangian multipliers λ, $\lambda_{1}$, and $\lambda_{2}$, and we form the Lagrange function.
(10)$$\begin{array}{cc} L(x,N,\lambda ) & =\phi \varepsilon l\left[r_{1}N-\eta \left(C+cN^{2}+dN\right)\right]+\left(1-\phi \right)\varepsilon l\left[r_{2}x+\xi r_{2}Z-\eta \left(A+\frac{e}{Z}x^{2}-ex+eZ\right)\right]\\ \; & \;\;\;\;\;\;\;\;\;\;\;-\lambda \left[r_{1}\phi \varepsilon lN+\left(1-\phi \right)\varepsilon l\left(r_{2}x+\xi r_{2}Z\right)-R_{b}\right]+\lambda_{1}N+\lambda_{2}x. \end{array}$$

We can find the optimal values for *x* and *N* by differentiating L(x,N,λ) with respect to *x* and *N* as follows. We define $L_{x}$ and $L_{N}$ as the partial derivative with respect to *x* and *N*, respectively.
(11)$$\begin{array}{cc} \; & L_{x}=2\left(1-\phi \right)\varepsilon l\left(r_{2}-2\eta \frac{e}{Z}x+\eta e\right)-\lambda \left(1-\phi \right)\varepsilon lr_{2}=0 \end{array}$$
(12)$$\begin{array}{cc} \; & L_{N}=\phi \varepsilon l\left(r_{1}-\eta 2cN-\eta d\right)-\lambda \phi \varepsilon lr_{1}=0 \end{array}$$
(13)$$\begin{array}{cc} \; & \lambda [r_{1}\phi \varepsilon lN+\left(1-\phi \right)\varepsilon l\left(r_{2}x+\xi r_{2}Z\right)-R_{b}]=0 \end{array}$$
(14)$$\begin{array}{cc} \; & \lambda_{1}N=0\;\lambda_{2}x=0, \end{array}$$
where $\lambda ,\lambda_{1},\lambda_{2}\geq 0$ and (13) is the complementary slackness condition. The objective function (9a) is a concave function with respect to *x* and *N*. Thus, the maximum can be obtained by the Karush Kuhn Tucker theorem [33]. We then analyse whether the constraints are binding. In (14), since N,x>0, so $\lambda_{1},\lambda_{2}=0$.

When λ>0, then the right hand part of (13) equals to zero. By setting Equations (11) and (12) equal to zero we can remove the λ. Then, we obtain the relationship of number of the verifiers *N* and computation resource *x*
$$r_{1}\left(r_{2}-2\eta \frac{e}{Z}x+\eta e\right)=r_{2}\left(r_{1}-2\eta cN-\eta d\right),$$
we substitute (Equation 15) into the constraint and then derive the optimal solutions
(15)$$\begin{array}{cc} N^{*} & =\frac{R^{\prime }-\frac{Zr_{2}\left(er_{1}+dr_{2}\right)}{2er_{1}}\left(1-\phi \right)\varepsilon l}{r_{1}\phi \varepsilon l+\frac{Zr_{2}c}{er_{1}}\left(1-\phi \right)\varepsilon l} \end{array}$$
(16)$$\begin{array}{cc} x^{*} & =\frac{Z}{2er_{1}}\left(er_{2}+r_{2}d+2cr_{2}N^{*}\right), \end{array}$$
where $R^{\prime }=R_{b}-\left(1-\phi \right)\varepsilon l\xi r_{2}Z$. To verify if λ>0 with the optimal value in (16), we substitute the optimal number of verifiers $N^{*}$ into (12). We obtain
(17)$$1-\frac{\eta }{r_{1}}d-\frac{2\eta cR^{\prime }-\frac{\eta cZr_{2}\left(er_{1}+dr_{2}\right)}{er_{1}}\left(1-\phi \right)\varepsilon l}{{r_{1}}^{2}\phi \varepsilon l+\frac{Zr_{2}c}{e}\left(1-\phi \right)\varepsilon l}=\lambda .$$
When (17) is positive, then λ>0, and the optimal value is obtained.

When λ=0, the constraint in (9b) is non-binding. Due to the fact that the objective function is concave, this means the reward budget is sufficient to the H-chain, which enables the BCAs to make the optimal contribution regardless of the reward budget. Hence, we can obtain the optimal values directly from Equations (11) and (12) which are
(18)$$\begin{array}{cc} N^{*} & =\frac{r_{1}-\eta d}{2\mu c} \end{array}$$
(19)$$\begin{array}{cc} x^{*} & =\frac{(r_{2}+\eta e)Z}{2e}. \end{array}$$
Up to this point, we obtain the optimal value of the number of the verifiers and the computation resource of the winning miner.

## 7. Simulation and Results

In this section, we demonstrate the results of the energy consumption with respect to different number of organisations, network settings, puzzle difficulty, and block size. In addition, we show the simulation results of the reward scheme of the proposed H-chain.

We first present the setting related to the blockchain. For PoW and PBFT, we assume that the complexity of one transaction is 20 bytes. And that there are [10,100] transactions in one block. Thus the blocksize is in the range of [0.2,2] KB [34]. The size of ACK message is about 20 bytes [35]. H-chain enables customisation of PoW since we are not planning to issue cryptocurrency. Thus, the organisation is able to define the PoW puzzle size according the applications. Hence, we set the basic puzzle complexity as 100 KB. The important parameters for the simulation are listed in Table 2.

Note that we assume the transmission power cost of a switch (Cisco Nexus 2224TP switch) (Cisco, San Jose, CA, USA) in the edge computing architecture is 48 W per port [36,37]. The computation power is related to the character of the computational task (i.e., computation strong, or I/O strong), and the CPU frequency. We assume that there are 4 BCAs (i.e., physical servers) which are used in the edge computing environment, each of which has 4-core Xeon (2 GHz) [38]. Therefore, the average computation power is 150 W. The evaluation was performed by implementing the analytical solutions using Python 3.8.

### 7.1. Energy Consumption of H-Chain

#### 7.1.1. Consensus within One Organisation

We first consider the simple scenario where all consensus happens within one organisation. Although, deploying PoW in a single organisation is unrealistic, for the sake of comparison we consider such deployment in order to observe the difference between PoW and PBFT energy consumption under the same setting. The energy consumption of PoW and PBFT in a single organisation is defined according to (Equation 3) and (Equation 4). In Figure 2, we consider the parameter values shown in Table 2. Note that the complexity difference between the puzzle of PoW and transaction verification of PBFT is $\gamma^{*}:\gamma^{\prime }=100:1$, and the difficulty of the puzzle is D=2. As the number of BCAs increases, both energy consumption of the PoW and PBFT increases. We, also, observe that the increasing of transactions in the block affects the energy consumption of both PoW and PBFT. Most importantly, PBFT in a single organisation shows greater advantage than PoW, which justifies that when the scale of the network is relatively small, deploying PoW is unnecessary for private information verification.

#### 7.1.2. Consensus across Organisations

In the inter-organisation scenario, we assume that the leading BCA of one organisation sends the newly built block to the other leading BCA of the other organisations. According to (Equation 3) and (Equation 4), we first study the influence of the inter-organisation extra cost factor $\beta (T_{p})$ to the energy consumption in Figure 3a. Then, we investigate the energy consumption with respect to the number of BCA verifiers under the same $\beta (T_{p})$ in Figure 3b.

In Figure 3a, we assume that there are 3 BCAs in each organisation and that the block contains 10 transactions. When the inter-organisation extra cost factor is relatively small, which indicates the links between organisations experience lower latency and communication cost, we can observe that PBFT has lower energy consumption per block. While, when the inter-organisation extra cost factor increases, the energy consumption of PBFT surpasses PoW. This is because PBFT is a communication intensive consensus mechanism that requires all-to-all communication amongst all the BCAs to be performed twice. Due to the transmission latency caused by inter-organisation links, the energy consumption of each block also increases with the number of organisations. The intersections in Figure 3 indicate that under the specific setting, the energy consumption per block of the PoW and the PBFT are equal to each other. This result serves as a guideline for the deployment of H-chain to the users.

The results are as expected. In particular, with increasing number of BCAs, the number of transactions per block, the inter-organisation extra cost factor, and the energy consumption per block increases for both PoW and PBFT. Therefore, we can conclude that by introducing H-chain, the organisations can be more flexible with customising their consensus mechanism. The results in this work also show that under specific network settings and consensus requirements, H-chain can enable the transactions to “go public” directly with reasonable energy consumption.

#### 7.1.3. Exploration of the Difficulty of the Puzzle

In Figure 4a, we simulate the impact of the PoW difficulty on the total energy consumption per block. When D=2, the difficulty of the puzzle is at a standard level. First, we study a scenario within the three organisation and five organisations, respectively. Since PoW is computationally intensive, the number of organisations affects the energy consumption only due to the inter-organisation extra cost factor. Second, according to (Equation 2) expressing the PoW’s difficulty and the puzzle size, the energy consumption of PoW increases dramatically with the difficulty of the puzzle. In Figure 4b, we evaluate total energy consumption per block of the different number of BCA miners in each organisation with respect to different difficulty factor. We can observe that both the number of BCA miners and the difficulty factor dominate the total energy consumption of PoW.

In Figure 5, we examine the energy consumption with respect to the blocksize. In this scenario, there are 5 BCAs in each organisation. First, when we compare PoW and PBFT in 3 organisations (which is the orange dotted line, blue dotted line, and the green line), we observe that PBFT has lower energy consumption per block. However, when we deploy PBFT across 5 organisations, the energy consumption surpasses PoW consensus, which also shows that under specific network and consensus requirements, it is better to utilise the public chain of H-chain instead of the private chain. Second, when the blocksize is relatively small, the advantage of PBFT is obvious. These simulations, which cover various requirements, allow the deployers of H-chain to form a clear idea of how best to utilise it to fit with their applications when considering energy consumption.

### 7.2. Evaluation of the Reward Plan

For the proposed reward plan in this paper, we first recall the optimal number of the verifiers in the PBFT and the optimal value of the winning miner’s utilised resource in the PoW. When the reward budget is unbounded, the social welfare is maximised through the optimised variables according to (18) and (19), where the rewards $r_{1}$ and $r_{2}$ to H-chain are proportional to the optimal number of BCA verifiers and the resource contribution of the winning miners. However, in reality, the reward budget is limited most of the time. Thus, we focus on the social welfare maximisation with limited reward budget, where the optimal optimisation variables are given by (15) and (16). From the above equations, we can see the optimal solution is dependent on the budget $R_{b}$, the reward to the verifier $r_{1}$, the reward to the miners $r_{2}$, and the proportion of the private blocks ε during *l* period.

First, we evaluate how the proportion of the private block affects the optimal values under limited budget. In Figure 6a, we have set different reward ratio, that is, $r_{2}=r_{1}$, $r_{2}=1.5r_{1}$, and $r_{2}=0.5r_{1}$, to stimulate the number of BCA verifiers in the private blockchain. In addition, we evaluate the resource contribution of the winning miner in the public chain with respect to the number of the miners in Figure 6b; here we have assumed that there is only one miner in each organisation, so the number of the organisations equals to the number of the miners. From Figure 6, we observe that as the proportion of the private blocks increases, the number of BCA verifiers and the resource contribution of the winning miner decrease. For the private chain, the cost increases as the private blocks increase. However, with a fixed reward of the increasing cost, the only way to maintain the profit is to reduce the number of verifiers of the private chain. The same reason holds for the resource contribution of the winning miner. We can also observe that when the reward $r_{1}$ to PBFT is bigger than $r_{2}$, there are more BCA verifiers in PBFT. In Figure 6b, we set the reward to PoW more than PBFT to stimulate the contribution of the winning miner under the limited reward budget. With the increasing number of the miners, the winning miner contributes more resource in solving the puzzle in order to obtain higher probability of winning, according to (Equation 6). In addition, when the reward $r_{1}$ to the BCA verifiers is fixed, a bigger reward ratio $\frac{r_{2}}{r_{1}}$ leads to more contribution of the winning miner.

We understand the ratio of the rewards to the private and public chain affects the value of the optimal solution, and furthermore the social welfare simultaneously. Thus, we study the relationship among the rewards $r_{1}$, $r_{2}$, and the social welfare in Figure 7a under the fixed proportion of the private block ϕ=0.5. Because of the nature of the optimisation problem with fixed reward budget, we observe that the highest social welfare is reached when the reward to the private chain is relatively small and the reward to the public chain is relatively big. This demonstrates that the cost of the PoW is higher than the PBFT, which also means the winning miner needs more reward to contribute its resource, so as to maximise the utility.

In Figure 7b, we investigate the impact of the budget to the H-chain. We set the reward budget from 8000 to 9000 with $r_{1}=r_{2}$. We observe the increase of the social welfare, which is due to the increasing number of verifiers and the resource contribution of the winning miner. For H-chain, it is clear that if the reward budget is sufficient, the H-chain can provide better verification for the applications by putting more resource towards it.

## 8. Conclusions

In this work, we proposed a hybrid blockchain, namely H-chain, to facilitate flexible information verification and validation of multiple organisations. H-chain aims to combine the advantage of both PoW and PBFT based consensus mechanisms. Further we proposed a novel architecture and the consensus mechanisms for H-chain. In addition, we design the reward plan to compensate the energy cost of H-chain, which also stimulates the BCAs to make the best effort for the consensus mechanism under fixed reward budget. We realise that, in the considered scenario, deployers will face challenges when choosing consensus mechanisms according to the energy consumption. Thus, we simulate different consensus settings and requirements, such as the blocksize, the number of transactions, the number of BCAs, the number of organisations, and the inter/intra-organisation transmission cost. The simulation results provide to the readers a clear picture of how to utilise H-chain in order to optimise energy consumption. For example, we show that the social welfare of H-chain varies significantly with the reward to the private chain, $r_{1}$, and the reward to the public chain $r_{2}$; specifically, the social welfare is maximised as the ratio $r_{2}/r_{1}$ exceeds seven. Conversely, when the reward is at the lowest for both the public and private blockchain the social welfare is at its lowest. This is not a surprising result but does indicate that the reward strategy is the key component towards optimising the performance of a hybrid blockchain.

## Figures and Tables

**Figure 1 sensors-21-00305-f001:**
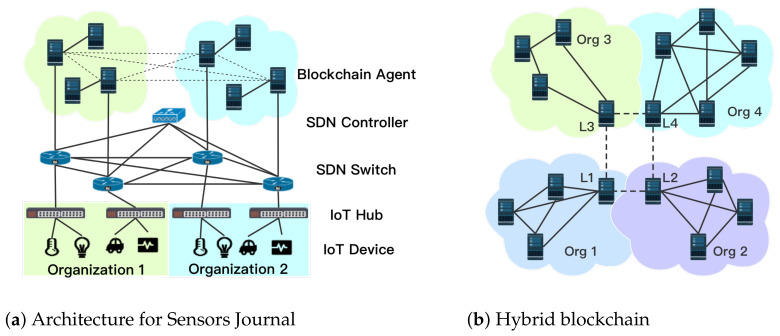
Illustration of scenario and hybrid-chain architecture.

**Figure 2 sensors-21-00305-f002:**
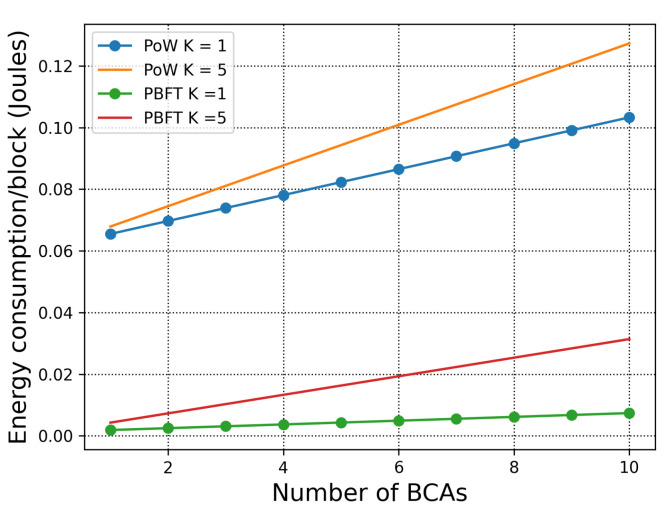
Total energy consumption per block with respect to the number of BCAs. $\gamma^{*}:\gamma^{\prime }=100:1$, D=2.

**Figure 3 sensors-21-00305-f003:**
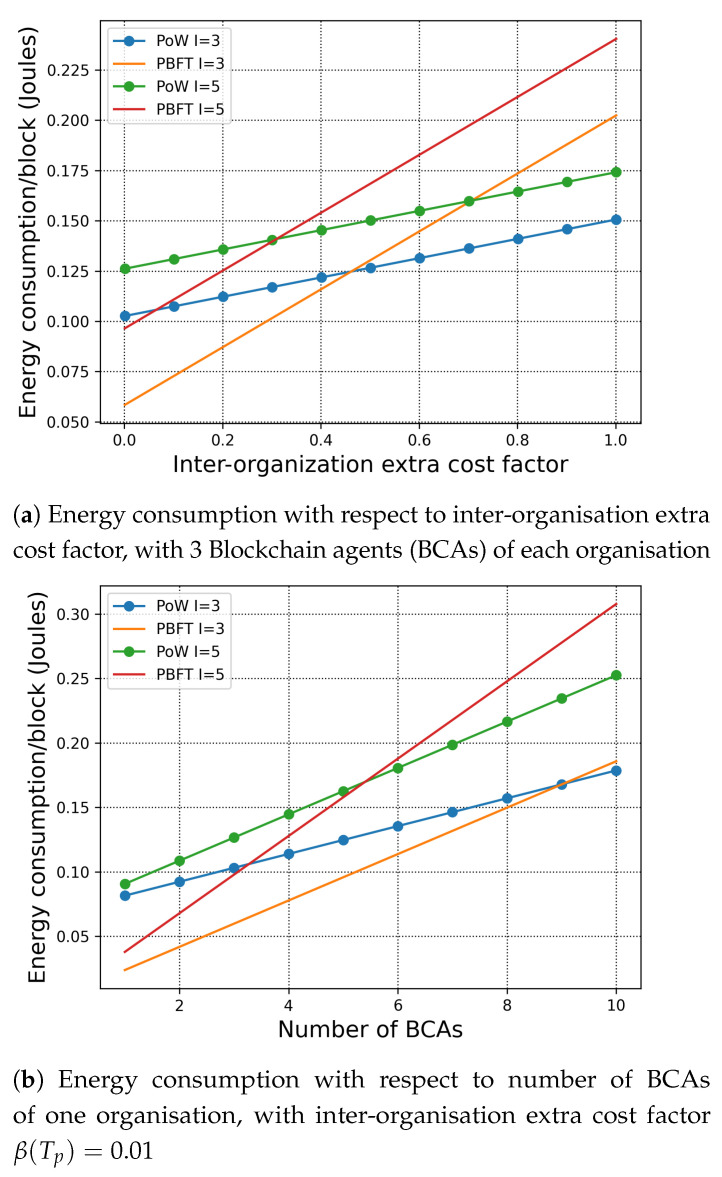
Total energy consumption with respect of the inter-organisation extra cost factor and the number of BCAs. $\gamma^{*}:\gamma^{\prime }=100:1$, D=2, K=10.

**Figure 4 sensors-21-00305-f004:**
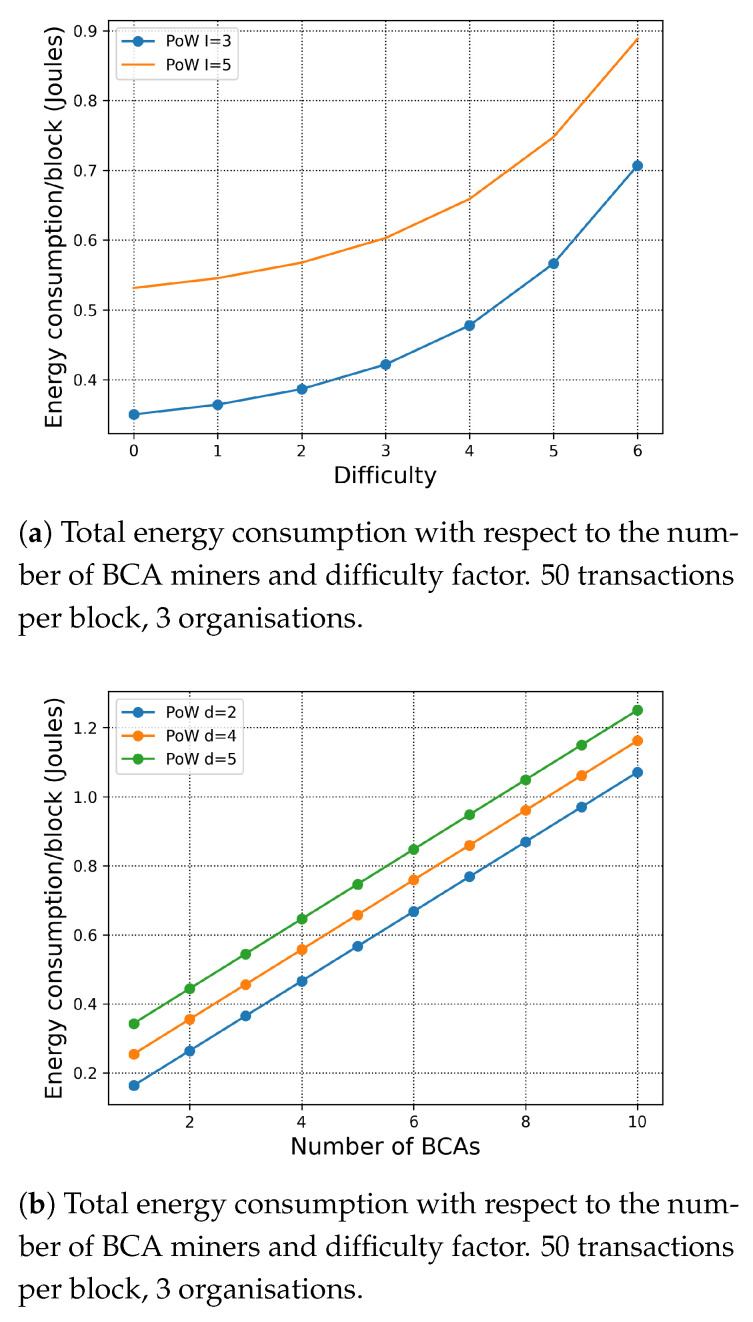
Total energy consumption with respect to the difficulty factor.

**Figure 5 sensors-21-00305-f005:**
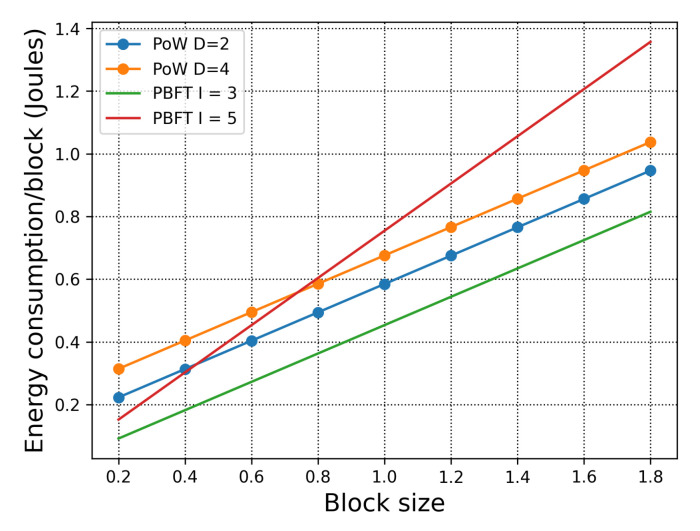
Total energy consumption with respect to the blocksize. With 5 BCAs in 3 organisations for Proof of Work (PoW), and 5 BCAs each organisation for Practical Byzantine Fault Tolerance (PBFT).

**Figure 6 sensors-21-00305-f006:**
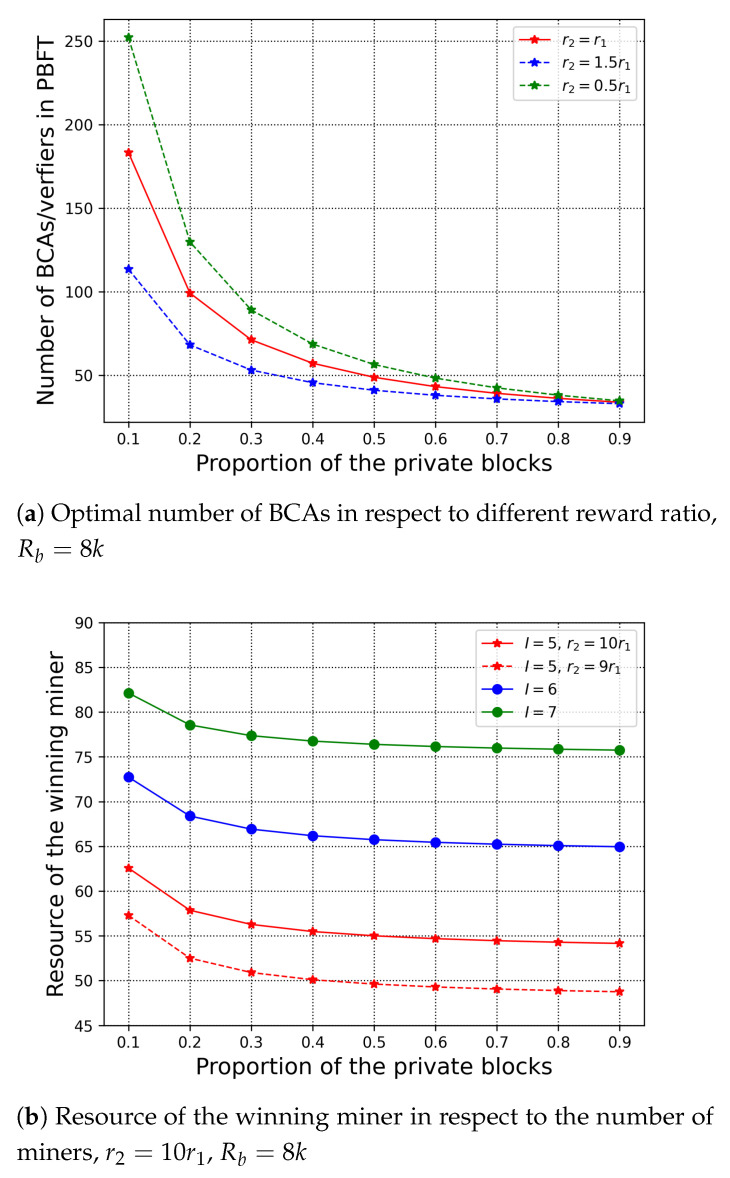
Proportion of the private blocks in respect to the number of BCAs and the resource of winning miner.

**Figure 7 sensors-21-00305-f007:**
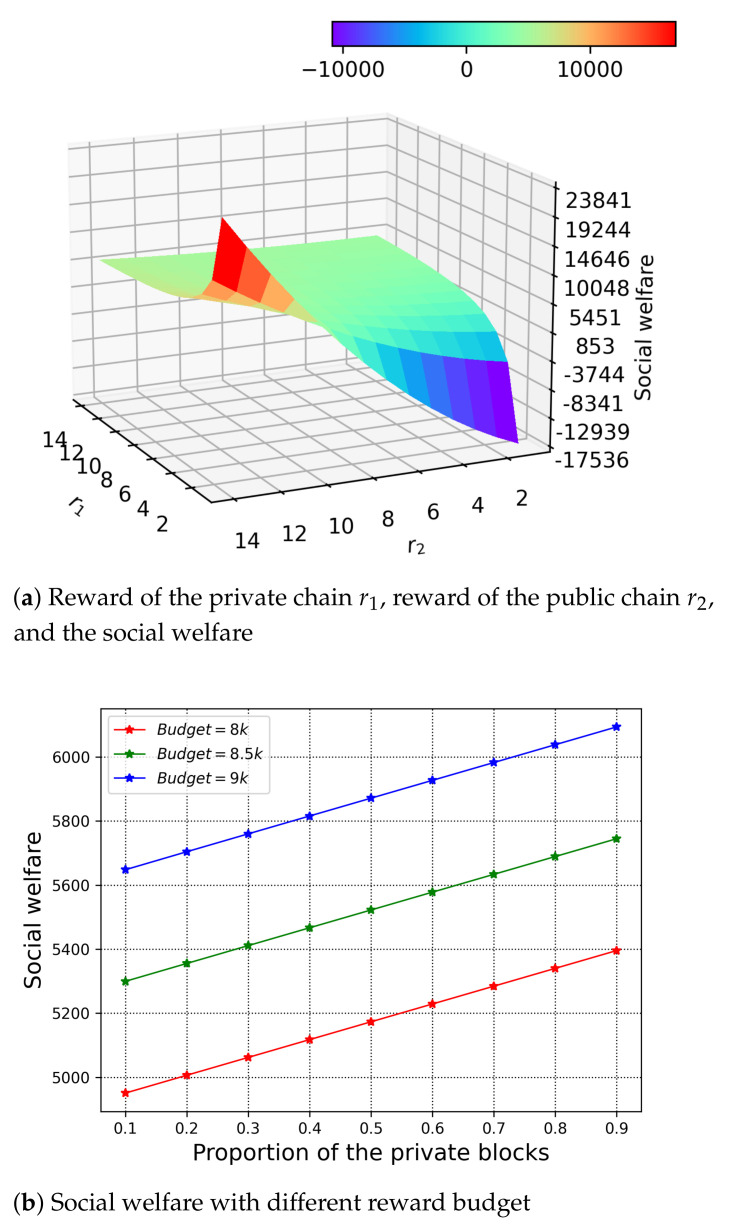
Social welfare and rewards.

**Table 1 sensors-21-00305-t001:** Notations and Descriptions.

Notation	Description	Notation	Description
$P_{t}$	Transmission power	$N_{i}$	Number of BCAs of Organisation *i*
$P_{c}$	Computation power	$C_{1}$	Computation latency of a new block
*R*	Intra-organisation effective throughput	$C_{2}$	Computation latency of winning miner
$R_{c}$	Inter-organisation effective throughput	$C_{3}$	Computation latency of new block verification
$T_{b}$	Intra-organisation dissemination latency	$\gamma^{\prime }$	Size of verification task
$T_{c}$	New block transmission latency	κ	Difficulty coefficient
β	Extra transmission cost factor	*B*	Basic puzzle size
*f*	CPU capability	*D*	Difficulty factor
*s*	Block size	ξ	Reward difference of the miners
*K*	Number of transactions in one block	ϕ	Proportion of the private blocks
$s_{ack}$	Size of ACK message	ε	Block rate
$\gamma^{*}$	Size of winning miner’s puzzle	*l*	Period
$\gamma_{i}$	Size of non-winning miner’s puzzle	$p_{l}$	Probability of being a winning miner
μ	Energy price per unit	$r_{1}$	Reward to the BCA verifiers
*I*	Number of organisations/miners	$r_{2}$	Reward to the winning miner
*N*	Number of BCAs	$R_{b}$	Reward budget

**Table 2 sensors-21-00305-t002:** Parameters and Value.

Parameters	Value	Parameters	Value
$P_{t}$	48 W	$\gamma^{\prime }$	1 KB
$P_{c}$	150 W	κ	0.1
*R*	10 Gbps	*B*	100 KB
$R_{c}$	1 Gbps	ξ	2
*f*	2 GHz	ϕ	ϕ∈(0,1)
*s*	0.2 KB–2 KB	ε	5
$s_{ack}$	20 Bytes	*l*	10 ms
$\gamma^{*}$	100 KB	$R_{b}$	8000
η	1		

## Data Availability

Data collected or analyzed in this study are not available for sharing.

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
