# Peer review of "Hybrid Blockchain for IoT—Energy Analysis and Reward Plan"

_sensors, 2021, doi:10.3390/s21010305_

Round 1
Reviewer 1 Report
Summary:
In the manuscript, the authors propose a hybrid-blockchain (H-chain) framework for scenarios where multiple organizations exist and where the framework allows private transaction verification and public transaction sharing and audit. In addition, the presented work studies the energy consumption of the hybrid consensus mechanisms in H-chain. Also, the authors propose a reward plan to incentivize the blockchain agents so that they make contributions to the H-chain while also considering the energy consumption. The manuscript illustrates the framework in a scenario that secures an Internet of Things (IoT) application connected via a software defined network (SDN). The results of simulation show that the reward plan incentivizes the blockchain agents to contribute to the H-chain considering the energy consumption of the hybrid consensus mechanism, enabling the proposed H-chain to achieve optimal social welfare.
In the manuscript presented, the authors address a highly interesting and very current topic. In addiction, the manuscript is well organized and structured, nevertheless I consider that the authors should attend to certain minor issues in order to improve the good quality of work.
Comments and Suggestions:
- Bitcoin should have its own reference just like Ethereum has it in the manuscript.
- Even if the reader knows the subject, the authors should expand the first occurrence of any the acronyms or initialism, some of them are not expanded as an example IP, SerIoT.
- Page 7 , Several definitions or notations are not included en Table 1 ( Notation and Descriptions) as K for number of transactions in one block, Computational latency C1, C2, intra-organisation dissemination latency Tb, etc. Including all the nomenclature and notations in Table 1 greatly helps the reader.
- Figure 5 and Figure 7, The two captions of the charts should be a little further apart because it can be confused as a single caption for both charts.
- Page 11, the method of Lagrangian relaxation should be including a reference.
- In section 7 (Simulation and results), the authors never clearly mention or refer to the software tools that were used to carry out the simulation process.
- Section 8: (Conclusion), the authors highlight a series of activities and results of the proposed hybrid blockchain together with its architecture and a consensus mechanisms making reference to the simulation results but without numbers way (qualitative). This section should be improved by the authors including numbers (quantitative) referring to the simulation results. In addition, it is also important to indicate the disadvantages and conditions or the worst scenarios in which the proposed hybrid blockchain together with its architecture and a consensus mechanisms could present failures.
.oOo.
Author Response
Dear Reviewer,
Please see the attached file.
Sincerely,
Dr. Jiejun Hu on behalf of the authors

Reviewer 2 Report
The paper is quite interesting since it addresses a known significant issue of Energy consumption associated with IoT and the relevant management issues based on Hybrid blockchain technology.
The related work section includes recent proposals and is a good background of the proposal.
The devised consensus mechanisms is interesting. The consequent reward plan that aims to encourage the number of checkers in the private chain and the relevant resource contribution of the miners in the public chain is well done. The proposal is well illustrated and the evaluation, based on simulation, is fair.
For what concerns the presentation, the authors could comment the derivation of equation (4), which is central in the proposal, and its sensitiveness to small fluctuation of paremeters that could happen in operation.
Author Response

(The authors gave the same response as above.)

Reviewer 3 Report
In this paper, the authors proposed hybrid blockchain for multiple organisation IoT applications. The hybrid blockchain aims to benefit from both the PoW-based and PBFT-based consensus mechanism. This paper first studied the energy consumption of H-chain by analytical approach and extensive simulation; Second, it designed a reward plan to incentivise the blockchain agents according to the energy consumption. This paper is technically sound and written clearly. However, there are a few points that I would like the authors to consider and revise.
- In Introduction section, the authors presented “three different types of knowledge” in line 50. However, in line 57, “Consequently, depending upon the type of information (private/shared/public),…”. Please justify the difference between knowledge and information.
- Line 59, “… we illustrate in Fig. 1(a)” should be “Fig. 1”. I suggest the authors refer Scenario 1 and 3 with Fig. 1
- Line 104 “To our knowledge, this is the first work that proposes hybrid-blockchain in SDN-based IoT applications to enable the flexible consensus mechanism and energy compensation based reward plan. …”. I understand that this work is an extension of the previous work. However, SDN-based IoT is weakly elucidated in the paper. The innovation of this work is to expand single organisation used private blockchain into H-chain aided multiple organisation cooperation with reward plan. SDN-based IoT could be one of the use case scenario to explain the operation of the H-chain.
- In Related work Section, the authors thoroughly reviewed hybrid blockchain and the energy consumption of the general blockchain technology. In subsection 2.1 “Hybrid blockchain”, I encourage the author consider “consortium blockchain” that is semi-decentralised and operated by multiple organisations. I also suggest the authors to distinguish “consortium blockchain”and “hybrid blockchain” by adding related works about consortium blockchain, just to name one, such as:
“Li, Zhetao, et al. "Consortium blockchain for secure energy trading in industrial internet of things." IEEE transactions on industrial informatics 14.8 (2017): 3690-3700.”
- In Section 5.1 Proof-of-Work and energy modelling, I believe the authors mean to say “permissioned-PoW”, since the workflow presented is not classed PoW. Moreover, in the workflow of permissioned-PoW Line 280 “the leading BCA decides the proportion of the blocks that goes public during a period of time. ”. I suggest the author indicate this point in the Introduction and contribution.
- In Section 5.2, Line 302, is the BCA(Initiator) the leading BCA in the organisation? Please be careful with the new terminology.
7. In Simulation section, Fig. 4, I encourage the authors interpret the intersections of the permissioned-PoW and PBFT under the same number of organisations.
Author Response

(The authors gave the same response as above.)

Round 2
Reviewer 3 Report
The author had answered all my questions well, so I suggest to accept this paper.